# Spray Drying of Blueberry Juice-Maltodextrin Mixtures: Evaluation of Processing Conditions on Content of Resveratrol

**DOI:** 10.3390/antiox8100437

**Published:** 2019-10-01

**Authors:** César Leyva-Porras, María Zenaida Saavedra-Leos, Elsa Cervantes-González, Patricia Aguirre-Bañuelos, Macrina B. Silva-Cázarez, Claudia Álvarez-Salas

**Affiliations:** 1Facultad de Ingeniería, Universidad Autónoma de San Luis Potosí, Av. Dr. Manuel Nava 6, San Luis Potosí 78210, Mexico; cesar.leyva@cimav.edu.mx (C.L.-P.); claudia.salas@uaslp.mx (C.Á.-S.); 2Centro de Investigación en Materiales Avanzados S.C. (CIMAV), Miguel de Cervantes # 120, Complejo Industrial Chihuahua, Chihuahua 31136, Mexico; 3Coordinación Académica Región Altiplano, Universidad Autónoma de San Luis Potosí, Carretera Cedral Km, 5+600 Ejido San José de las Trojes, Matehuala 78700, Mexico; elsa.cervantes@uaslp.mx (E.C.-G.); macrina.silva@uaslp.mx (M.B.S.-C.); 4Facultad de Ciencias Químicas, Universidad Autónoma de San Luis Potosí, Av. Dr. Manuel Nava 6, San Luis Potosí 78210, Mexico; paguirreb@uaslp.mx

**Keywords:** resveratrol, analysis of variance (ANOVA), spray drying, blueberry juice-maltodextrins, conservation of antioxidants

## Abstract

Resveratrol is an antioxidant abundant in red fruits, and one of the most powerful inhibiting reactive oxygen species (ROS) and oxidative stress (OS) produced by human metabolism. The effect of the spray drying processing conditions of blueberry juice (BJ) and maltodextrin (MX) mixtures was studied on content and retention of resveratrol. Quantitatively, analysis of variance (ANOVA) showed that concentration of MX was the main variable influencing content of resveratrol. Response surface plots (RSP) confirmed the application limits of maltodextrins based on their molecular weight, where low molecular weight MXs showed a better performance as carrying agents. After qualitatively comparing results for resveratrol against those reported for a larger antioxidant molecule (quercetin 3-D-galactoside), it was observed a higher influence of the number of active sites available for the chemical interactions, instead of stearic hindrance effects.

## 1. Introduction

Human metabolism continuously produces reactive oxygen species (ROS) by normal respiration and cellular functions. These species may cause cellular damage by affecting the DNA, stimulating free radical chain reactions, and provoking more than one hundred diseases [1]. To counteract the reactive oxygen species (ROS), human body produces antioxidant compounds that may scavenge free radicals either by transferring free electrons or hydrogen atoms [2]. However, an over production of oxidative species and a weak immune system, may result in an imbalance in the body, developing oxidative stress (OS). This OS can generate changes in cell volume, and make other biomolecules (i.e., proteins, lipids and nucleic acids) to malfunction, leading to other major degenerative diseases such as cancer, diabetes, atherosclerosis, stroke, asthma, arthritis, dermatitis and aging, among others [3]. Thus, one approach to reduce OS is a regular consumption of antioxidants, which are found in fruits, vegetables, leafs and roots. There are various types of antioxidants in nature, among them are found flavonoids and stilbenoids structures such as quercetin 3-D-galactoside, resveratrol, myricetin and kaempferol [4,5]. Among these compounds, resveratrol (3,4’,5-trihydroxystilbene) is a polyphenolic compound that exists in the *cis*- and *trans*- isomeric forms, synthesized by plants as a phytoalexin in response to injury, fungal attack and exposure to UV light [6]. It is one of the most potent antioxidants against ROS and OS, and its intake is beneficial for human health, in the modulation of vascular cell function, suppression of platelet aggregation, reducing myocardial damage, inhibiting kinase activity, as anti-inflammatory and effective against the carcinogenesis [1,3]. Particularly, resveratrol is abundant in red wine, grape berry skins and seeds, berries, nuts and roots such as the Itadori plant (*Polygonum cuspidatum*) [7,8]. According to Tomé-Carneiro et al. the cardiovascular risk factor is reduced with a dose of 8 mg/day of resveratrol for one year [9]. In terms of red wine consumption, this dose would be equivalent to drinking 1–3 L of wine per day, depending on wine variety [10]. Additionally to a large volume ingestion of antioxidants containing products, utilization of resveratrol in the food industry is limited by factors such as low stability against oxidation, high photosensitivity, insolubility in water and short biological half-life (i.e., rapid metabolism and elimination) [11,12]. Thus, it is necessary to develop food products containing a high concentration of resveratrol, offering a reduced volume, while keeping the stability and bioavailability of antioxidants.

Both, flavor and high content of antioxidants in blueberry (*Vaccinium corymbosum*) juice (BJ) has triggered its consumption and popularity in regions such as Europe, Asia, North America and Latin America, but the short harvest season of the fruit and rapid perishability, limit its availability in the market [13]. In consequence, more than 50% of the total production must be processed in food products that may support a long shelf life such as juices, nectars, yogurts, marmalades, syrups and juice powders. Unfortunately, thermal degradation of antioxidants may occur during processing at temperatures higher than 60 °C. Spray drying of fruit juices is an alternative that offers a solution to this issue. In this sense, several studies have reported spray drying of different fruit juices with carrying agents. These agents are employed as aids in the drying process because of their relative high glass transition temperature (T_g_). Fruit juices such as orange, pineapple, apple, watermelon, mango and blueberry, have been successfully dried with carrying agents such as inulin, maltodextrins, starch, Arabic gum and sodium alginate [14,15,16,17,18,19,20]. Maltodextrins are polysaccharides obtained from the acidic hydrolysis of starch, with a nutritional contribution of only 4 calories per gram. Besides, maltodextrins are commercially found in a wide range of molecular weight distributions (MWD), which lead to different thermal properties and potential applications [21]. Recently, Saavedra-Leos et al. used a set of four maltodextrins as carrying agents in spray drying of BJ [22]. Through analysis of variance (ANOVA) and response surface plots (RSP), they determined the effect of processing conditions, found an optimal set of experimental variables for drying the juice and conserving of quercetin 3-D-galactoside, and set the application limits of maltodextrins based on their MWD. Although there is a large number of works published on the subject of conservation of resveratrol, few have addressed the issue of optimization of processing conditions. For example, Lim, Ma and Dolan carried out a systematic experiment for testing the spray drying conditions of blueberry by-products. They found a yield of 94% when employing a ratio of blueberry solid to maltodextrin of 30:70, and an outlet temperature of 90 °C [19]. Jiménez-Aguilar et al. studied the spray drying conditions of blueberry and mesquite gum on color and degradation of anthocyanins, and found a direct relation between color and concentration of anthocyanins [23]. Tatar Turan et al. studied the effect of an ultrasonic nozzle on spray drying of blueberry juice with maltodextrin and Arabic gum as carrier agent and coating material, respectively [24]. They reported the optimal drying conditions as inlet temperature of 125 °C, ultrasonic power of 9 W, and feed pump rate of 8%. Correia et al. encapsulated the polyphenolics in wild blueberry with different protein-food ingredients, by two-way ANOVA and Tukey’s test; they found that soy protein produced the highest total phenolic content [25]. Darniadi, Ho and Murray, also used ANOVA for comparing two methodologies in drying of blueberry juice: Foam-mat freeze-drying (FMFD) and spray drying. They concluded that the highest powder yield was achieved with FMFD methodology and a ratio of maltodextrin to whey protein of 1.5 [26]. Shimojo et al. employed a 2^2^ full factorial experimental design for evaluating the process parameters on the properties of resveratrol loaded in nanostructured lipid carriers [27]. Additionally, other works have also studied the drying processing conditions of alternative drying methodologies [28,29,30].

Therefore, in the present work, ANOVA and RSP were employed for setting the optimal processing conditions of blueberry juice-maltodextrin (BJ-MX) mixtures. The effects of inlet temperature, concentration and type of maltodextrin, were studied on content, and retention of resveratrol, and the results were compared with those reported by Saavedra-Leos et al. (2019) for quercetin 3-D-galactoside. This work contributes to understanding the application limits of maltodextrins when used as carrying agents in the spray drying of sugar-rich systems, and for conservation of antioxidants.

## 2. Materials and Methods 

### 2.1. Materials

Blueberry juice (BJ) was prepared using crushed fresh blueberry fruit (*Vaccinium corymbosum*), commercially available in a local market center (Costco Wholesale Corp., San Luis Potosí, Mexico). Prior to juice extraction, the fruits were stored in a refrigerator by 12 h, and crushed in a juice extractor Turmix E-17 (Guadalajara, Mexico). Juice and bagasse were stored in a glass container inside the fridge by 12 h, and after this period, were separated by vacuum filtration with paper filter Whatman No. 4, used in clarification of juices and wines. BJ was stored in darkness inside a refrigerator at 4 °C for avoiding degradation of antioxidants. 

Four types of maltodextrins (MX) were employed as carrying agents, and identified according to their dextrose equivalent (DE) as DE of 7 (commercial grade maltodextrin, CM), DE of 10 (M10), DE of 20 (M20) and DE of 40 (M40). CM dry powder was purchased from INAMALT (Guadalajara, Mexico), while M10, M20 and M40 from INGREDION Mexico (Guadalajara, Mexico).

### 2.2. Experimental Design

Without needing many experimental runs, and keeping a confidence interval [31], D-optimal experimental design ensures an optimal selection of spray drying conditions for maximizing the content of antioxidants. Then, for this purpose, two independent continuous variables (inlet temperature (T) and maltodextrin concentration (C)), and one categorical independent variable (type of MX) were tested. The minimum and maximum levels of these variables are described in Table 1. Content of resveratrol was set as unique response variable. The D-optimal experimental design consisted of 25 experiments, necessary to achieve a quadratic model in the quantitative factors. From these 25 experiments, five runs were repeated, i.e., runs 8, 10, 11, 12 and 14 were repeated with runs 21, 22, 23, 24 and 25, respectively. All the experiments were executed randomly. Experimental design and RSP were carried out employing Matlab 2013a (Natick, MA, USA).

### 2.3. Spray Drying

Dehydration of blueberry juice was carried out in a Mini Spray Dryer B290 (Buchi, Switzerland), by feeding mixtures of blueberry juice and maltodextrin (BJ-MX) into the spray dryer at room temperature. Blends of BJ-MX were prepared by adding the necessary amount of maltodextrin in the juice, and by constant mechanical stirring. Hot air was employed as drying vehicle, at a volumetric flow rate of 35 m^3^/h, and constant pressure of 1.5 bar. The rest of processing conditions were varied according to the testing level of variables described above.

### 2.4. Content and Retention of Resveratrol

Quantification of antioxidant content in dry samples was carried out by dissolving 0.5 g of powder in 0.5 mL of phosphoric acid (10% *v/v* in water), and 3 mL of methanol used as an extracting solvent. In order to maximize the extraction of antioxidants, the solution was stirred for 5 min, and left resting 24 h in darkness. Solution was filtered in an Acrodisc filter (0.45 μm), and the filtered was diluted with 200 μL of methanol. A constant volume of 10 μL was injected in a high performance liquid chromatography (HPLC) instrument. Injections were carried out by triplicate. Content of resveratrol was quantified by HPLC with a Waters system (Waters Assoc. Milford, MA, USA), equipped with a binary pump, an auto-injector (model 717), and a dual wavelength absorbance detector (model 2487). The analyses were carried out at room temperature, and a pH of 3.0 in the solution. A constant flow rate of 1 mL/min solution of 50% acetonitrile-phosphoric acid was employed as the mobile phase. Detection was set at a wavelength of 306 nm. Chromatographic separation was done with an Agilent Zorbax C-18 column (75 mm × 4.6 mm DI 3.5 μm). All data were analyzed with the Empower Pro software Version 4.0 (Mildford, MA, USA).

Calibration curves were constructed employing a resveratrol HPLC grade standard (>99.0%, Sigma-Aldrich, Toluca, Mexico). A stock solution of 1000 μg/mL, and several aliquots (0.01, 1, 5, 10 and 20 μg/mL) were prepared as the calibration curve. Calibration curves were prepared the same day of injecting in the HPLC. Elution time of resveratrol was 2.5 min, while mobile phase eluted at 0.92 min. The intensity of resveratrol peak linearly increased with concentration. Content of resveratrol in the sample was determined from comparison with the calibration curve. 

Resveratrol was evaluated in the original juice, and in dry powders. The content of antioxidants was expressed as micrograms of resveratrol per gram of blueberry juice powder (μg/g). Percent of retention (R) of resveratrol was determined according to equation (1):(1)$$R\;\left(\%\right)=\frac{Q_{P}\times 100}{Q_{j}},$$
where Q_P_ is the content of resveratrol in dry powder (in ppm), and Q_J_ is the content of antioxidants in fresh juice (8.38 ppm). 

### 2.5. Statistical Analysis

The effects of experimental variables (factors) and their interactions were evaluated with an analysis of variance (ANOVA). In ANOVA, the *F*-value indicates the effect of the independent variable on the response variable. Thus, if F is equal to one, the independent variable has no effect, while if *F* > 1 the independent variable has an effect and its effect is lager as *F* increases above 1. The *p*-value represents the probability of an *F*-value large enough for influencing the experiment; if this value is equal or lower than the significance level, then the assumption of the influence of the independent variables on the response variable is correct. The probability of rejecting the previous assumption even when it is true, is given by the significance level.

A quadratic model with second-order interactions and main effects were used to explain a relationship between the given continuous variables as indicated in Equation (2):Z = α_0_ + Σα_i_X_i_ + Σα_ii_X_ii_^2^ + Σα_ij_X_i_X_j_,(2)
where, Z represents the response variable (content of resveratrol), X_i_ and X_j_ are the factors (temperature, and concentration of maltodextrin) and α_0_, α_i_, α_ii_ and α_ij_ are the linear regression coefficients of the model. 

In the process of selecting a model, some parameters of the complete model were first adjusted with Equation (2). Based on a normality test of Anderson Darling for the response variable (R), a transformation of the response was made by the Box-Cox analysis when it was necessary to stabilize the variance. Then, for simplification, the model was hierarchically pruned, and used only with significant factors. Here we present the results obtained with the pruned model, and transformed into response variables.

## 3. Results and Discussion

### 3.1. Content and Retention of Resveratrol

From 25 experiments, 18 runs were successfully spray-dried, while in seven runs the microstructure collapsed. Since the results for resveratrol were discussed herein, results of physicochemical characterization and yield can be consulted in the work previously reported for quercetin 3-D-galactoside [22], since both works are parts of the same experiment. Table 2 shows the detailed experimental description of the 25 runs and results of content, and retention (R) of resveratrol at different processing conditions. Minimum and maximum values for content, and retention, of 18 spray-dried samples were 0.0–0.47 μg/g, and 0.0–10.24%, respectively. The corresponding average values for these two parameters were 0.28 μg/g, and 6.25%. The highest retention was obtained for run 10 with a value of 10.24%. Overall, average concentration of resveratrol in dried experiments diminished about 96%. This was calculated considering the initial concentration of resveratrol in fresh BJ, and the content of antioxidant determined in dried samples. On the other hand, content and retention values were relatively lower than those obtained for quercetin 3-D-galactoside [22]. Therefore, based on these observations, it is evident that resveratrol is less prone to interact chemically with maltodextrins, being more exposed to thermal degradation during the spray drying process. While some qualitative relations among the independent and categorical variables may be inferred from Table 2, the quantitative analysis of the effect of processing variables on content of resveratrol will be discussed in following section.

### 3.2. ANOVA and Response Surface Plots Analysis (RSP)

Table 3 shows ANOVA results calculated for the content of resveratrol. ANOVA results showed that concentration (C) of MX was the variable with the most important effect, while the type of maltodextrin, and inlet temperature had a negligible effect. These observations were confirmed by the *p*-value at a significance level of 0.05. The interactions between the same variable (intra) showed that only the concentration (C^2^) had an effect on content, but 2.4 times less than the single concentration (C). The rest of interactions between variables (inter and intra) such as T·C, T·MX, C·MX and T^2^ showed a *p*-value higher than the significance level, thus their effects were negligible. Therefore, concentration was the independent variable with the larger effect on the content of resveratrol. 

These observations indicated that concentration was the main processing variable affecting the content of resveratrol, while the type of maltodextrin and inlet temperature showed no effect on the response variables, suggesting that content of resveratrol was unresponsive to inlet temperatures exerted in the experimental design, and to differences in molecular weight distribution of MXs. 

Figure 1 shows RSP for the content of resveratrol as a function of the type of MX. The surfaces shape was similar regardless of the type of MX. In all cases, the highest content of resveratrol was observed at a concentration of MX of 23%. At higher or lower concentration values, content of resveratrol decreased rapidly. However, slight differences were observed with inlet temperature. CM showed a maximum content of resveratrol at 170 °C, and a decrement at 210 °C. M10 showed the opposite behavior, with a maximum value at 210 °C and a decrement at 170 °C. High molecular weight maltodextrins (M20 and M40) showed two maximum values at opposite temperatures. All of these observations allowed setting the optimal processing conditions for spray drying of BJ-MX. Highest values for content of resveratrol were obtained at a concentration of MX of 23%, and inlet temperature of 170 °C for CM or 210 °C for M10. In addition, RSP results confirmed the utilization limits of MXs, where low molecular weight MXs produced a higher content of resveratrol than high molecular weight MXs. These observations demonstrated that depending on the molecular weight distributions of MXs, these polysaccharides might be employed selectively as carrying agents in spray drying of diverse sugar-rich systems. Indeed, chemical interactions between maltodextrins and antioxidants (i.e., resveratrol) may be affected by other variables rather than by the degree of polymerization of maltodextrin. Other variables that may affect this interaction into a greater or lesser extent are the volumetric flow of solution injected into the dryer, wet bulb temperature (i.e., relative humidity of air), type of nozzle used (for example, regular versus ultrasonic) and adjuvant agents (i.e., soy protein and sodium alginate). However, in the present work, ANOVA was focused on two spray drying variables (concentration of maltodextrin and inlet temperature) and one categorical variable (type of maltodextrin). In this sense, Ameri and Maa, indicated that increasing the total content of solids in feed solution, increased the recovering of powders in spray drying [32]. Nadeem et al. concluded that yield was related to concentration of maltodextrin, rather than to drying temperature [33]. Caliskan and Gulsah found that increasing the concentration of maltodextrin resulted in an increment in the yield of dried powder [34]. Bhusari et al. attributed this behavior to an increment in the T_g_ of mixtures [35]. Peng et al. indicated that above 30% of the carrying agent was detrimental for product quality [36]. Saavedra-Leos et al. determined an inverse relation between T_g_ and DE of maltodextrins [21]. Through several works reported by the Saavedra-Leos group [14,22,37,38,39], we have observed that in some juices such as orange and blueberries, their physicochemical properties varied with the differences in the type and distribution of molecular weight of the carrier agent. In these studies, we have attributed this behavior mainly to: (i) Differences in the molecular weight distribution of the carrier agent, (ii) the arrangement of polymer chains (i.e., entangled or linear) and (iii) the type of molecule in these chains (for example, glucose for maltodextrin or fructose for inulin). Additionally, according to Darniadi, Ho and Murray, when mixing low molecular weight sugars and high molecular active compounds, the active compound tends to segregate in some extent to the surface of dried particles [26]. While this behavior prevents the particles sticking on the dryer walls, it also exposes the active ingredient to a faster degradation.

Table 4 shows the predictive equations for the content of resveratrol as a function of the type of MX. These equations were extrapolated from SRP and in consequence were only valid within the interval of conditions tested herein. Inlet temperature presented a negative effect and concentration a positive effect. From both processing variables, the extent of concentration was about 1.9 times larger than that of temperature. The values for interactions (T·C), and square of temperature (T^2^) showed a relatively low positive value, but their numerical contributions were similar to that of concentration. Although the square of concentration (C^2^) showed a negative effect, its value was still lower than that of single concentration, thus indicating a little contribution on the content of resveratrol. Interactions (T·C, T^2^ and C^2^) showed a constant value indicating that these interactions were insensitive with respect to the type of MX. Numerical calculations employing these equations supported the results found from RSP. CM showed the largest content value when using a temperature of 170 °C, while for M10 the higher content was obtained with a temperature of 210 °C. 

### 3.3. Effect of the Chemical Structure in the Content of Antioxidants

In this section we compared the results of the content of resveratrol reported herein, against those for quercetin 3-D-galacoside previously reported [22]. The effect of spray drying processing conditions showed similar behavior for both antioxidants, but the main difference relied on the values of the content of each antioxidant. In general, the content of quercetin 3-D-galactoside was 2–8 times higher than that of resveratrol. Conversely, according to Shrikanta, Kumar and Govindaswamy, the total content of polyphenols was 1–14 times higher than the total content of flavonoids in underutilized Indian fruits [8]. In this sense, Araujo-Díaz et al. reported a concentration of resveratrol 1.5 times higher than for 3-D-galactoside when employing maltodextrin in spray drying of blueberry juice [37]. Figure 2 shows a schematic representation of the chemical structure of both antioxidants, and a maltodextrin repetitive unit. Resveratrol is a stilbene with a C6-C2-C6 structure and three hydroxyls (OH^−^). While quercetin is a flavonoid with a C6-C3-C6 structure, five hydroxyls, one alkoxy group (ether) and one carbonyl group (ketone); the galactoside refers to the galactose molecule containing four hydroxyls. On the other hand, maltodextrins are polysaccharide molecules consisting of glucose units linked by glycosidic α-(1-4) and α-(1-6) bonds, with a variable length of its polymeric chains expressed as DE [40]. The glucose molecule contains three hydroxyl groups, one alkoxy group (ether) and one hydroxyl group at each extreme of polymeric chains. All these functional groups are responsible for carrying out molecular chemical interactions such as hydrogen bonding, and Van der Waals interactions. In several works it has been reported that these chemical interactions (inter and intramolecular) are responsible of the adsorption of water on different carrying agents such as inulin and maltodextrins [21,38]. In these works, water adsorption was promoted with increasing the molecular weight of carrying agents. The set of maltodextrins employed as carrying agents in this work, was similar to that reported by Saavedra et al. and presented a degree of polymerization (DP) of: CM 2-12, M10 2-16, M20 2-21 and M40 2-30, units of glucose [21]. Although a high DP may indicate a larger number of hydroxyl groups exposed for chemical interactions, the polymeric chains in MXs may arrange in different configurations rather than linearly, forming entangled branches, thus reducing the availability of active sites. Additionally to the arrangement of polymeric chains, the stearic hindrance between the adsorbing molecules and MXs, is another aspect affecting the final content of antioxidants. Evidently, the size of quercetin 3-D-galactosie molecule is larger than that of resveratrol, suggesting that stearic hindrance was not the main factor influencing larger chemical interactions with MX, but the number of functional groups available such as hydroxyls, alkoxys and carbonyls. Based on these arguments, it is possible to infer that: (i) The availability of functional groups is the main cause of chemical interactions, since quercetin has more of these groups than resveratrol; hence its greater interaction with maltodextrin, and (ii) in maltodextrins, the availability of these functional groups is limited by branching and entangled of polymeric chains. For this reason, in both cases i.e., the content of resveratrol and quercetin, low molecular weight MXs presented higher antioxidant content than high molecular weight MXs, indicating that polymeric chains in these carrying agents are less branched and entangled, thus promoting more chemical interactions.

## 4. Conclusions

The effect of processing conditions in spray drying of blueberry juice and maltodextrin mixtures (BJ-MX) on content and retention of resveratrol was studied. Analysis of variance (ANOVA) showed that concentration (C) was the main variable influencing the content of resveratrol. Response surface plots (RSP) allowed observing that the content of resveratrol was also affected by the molecular weight distribution of MXs employed as carrying agents, where low molecular weight MX presented a higher content of resveratrol. With this study, it was possible to set the optimal processing conditions for spray drying of BJ-MX such as concentration of 23% of maltodextrin and temperature of 170 °C for CM, and 210 °C for M10. Additionally, the results reported herein were compared with those reported for content of quercetin 3-Q-galactoside, finding that quercetin 3-Q-galactoside was more readily to interact with MXs because of a higher availability of functional groups rather than by stearic hindrance effects.

## Figures and Tables

**Figure 1 antioxidants-08-00437-f001:**
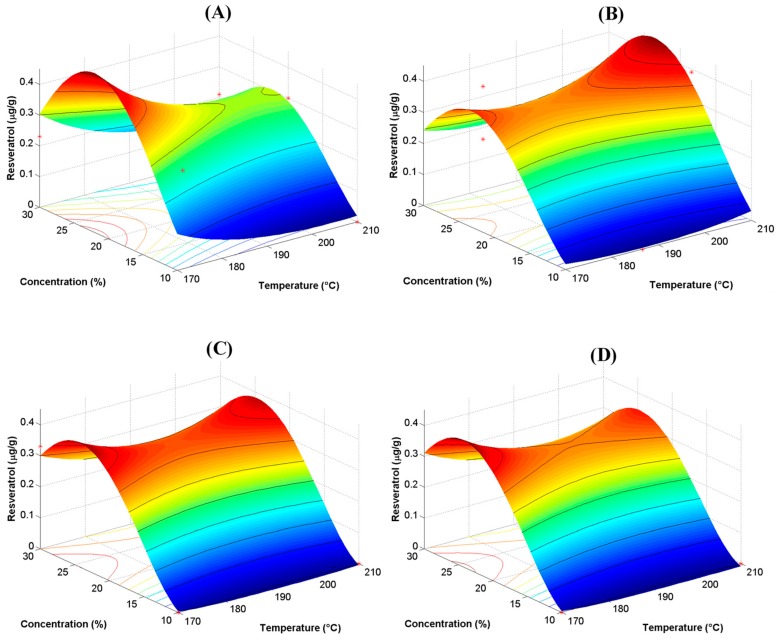
Response surface plots analysis (RSP) for the content of resveratrol as a function of the type of MX: (**A**) CM, (**B**) M10, (**C**) M20 and (**D**) M40. The symbols (*) in each RSP represent the center and start points for estimation of second order effects.

**Figure 2 antioxidants-08-00437-f002:**
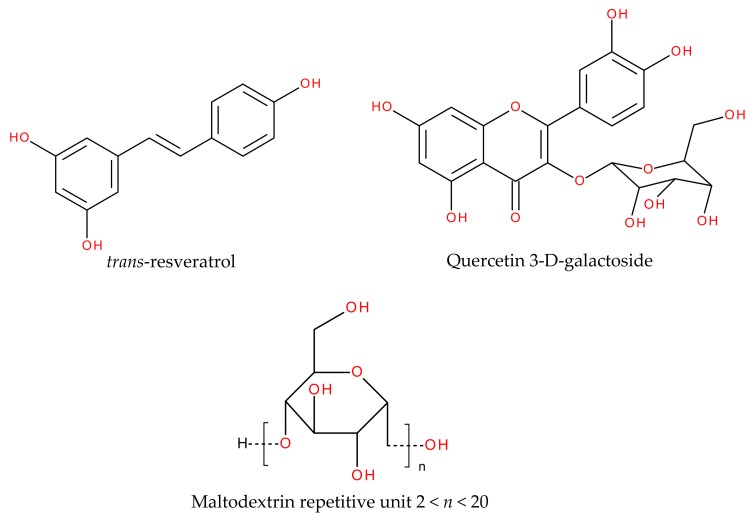
Schematic representation of chemical structures of resveratrol, quercetin 3-D-galactoside and the maltodextrin repetitive unit.

**Table 1 antioxidants-08-00437-t001:** Testing levels of variables in D-optimal experimental design.

Tested Variables	Testing Level	
	Minimum	Maximum
Inlet temperature (°C)	170	210
Maltodextrin concentration (wt%)	10	30
Dextrose equivalent	CM	M40

**Table 2 antioxidants-08-00437-t002:** Content and retention of resveratrol in spray drying of blueberry juice-maltodextrin (BJ-MX).

Run Identification	Factors			Resveratrol	
	T (°C)	C (wt%)	MD	Content (μg/g) ^a^	Retention (wt%)
1	170	30	CM	0.23 ± 0.031	5.01
2	187	10	M10	0	0.00
3	170	30	M20	0.33 ± 0.108	7.19
4	210	10	M40	0	0.00
5	210	30	CM	0.21 ± 0.056	4.58
6	210	18	M10	0.39 ± 0.036	8.50
7	210	10	M20	0	0.00
8	170	10	M40	0	0.00
9	210	10	CM	0	0.00
10	170	22	M10	0.47 ± 0.282	10.24
11	170	10	M20	0	0.00
12	181	30	M40	0.29 ± 0.010	6.32
13	190	20	CM	0.29 ± 0.020	6.32
14	193	30	M10	0.22 ± 0.053	4.79
15	210	30	M20	0.27 ± 0.122	5.88
16	175	12.5	CM	0.28 ± 0.020	6.10
17	190	20	M10	0.28 ± 0.212	6.10
18	190	25	M20	0.31 ± 0.057	6.76
19	209	24	M40	0.26 ± 0.033	5.67
20	210	20	CM	0.30 ± 0.065	6.54
21	170	10	M40	0	0.00
22	193	30	M10	0.20 ± 0.019	4.36
23	170	10	M20	0	0.00
24	181	30	M40	0.25 ± 0.020	5.45
25	170	22	M10	0.3 ± 0.170	6.54

^a^ Average and standard deviation values calculated from three repetitions.

**Table 3 antioxidants-08-00437-t003:** ANOVA results determined for content of antioxidant in BJ-MX.

	Content				
Source	DF	SS ^a^	MS ^b^	F	*p* *
Model	14	1.069	0.076	15.978	0.0001
T	1	0.011	0.011	2.425	0.150
C	1	0.531	0.531	111.03	0.0001
MX	3	0.005	0.002	0.401	0.755
T·C	1	0.0005	0.0005	0.122	0.733
T·MX	3	0.021	0.007	1.516	0.269
C·MX	3	0.033	0.011	2.308	0.138
T^2^	1	0.011	0.011	2.463	0.147
C^2^	1	0.222	0.222	46.605	0.0001
Residual	10	0.047	0.004		
Total	24	1.117			

^a^ Sum of squares; ^b^ mean squares; * calculated at a significance level of 0.05; T = inlet temperature, C = maltodextrin concentration, MX = type of maltodextrin.

**Table 4 antioxidants-08-00437-t004:** Predicting equations extrapolated from SRP for the content of resveratrol in BJ-MX, as a function of type of MX.

Type Of MX	Content
**CM**	$\ln\left(Q+0.01\right)=5.8694-0.0639T+0.1069C+0.00003T\cdot C+0.00015T^{2}-0.0025C^{2}$
**M10**	$\ln\left(Q+0.01\right)=4.6662-0.0583T+0.1137C+0.00003T\cdot C+0.00015T^{2}-0.0025C^{2}$
**M20**	$\ln\left(Q+0.01\right)=4.8695-0.0602T+0.1191C+0.00003T\cdot C+0.00015T^{2}-0.0025C^{2}$
**M40**	$\ln\left(Q+0.01\right)=5.0389-0.0611T+0.1193C+0.00003T\cdot C+0.00015T^{2}-0.0025C^{2}$

Q = content of resveratrol (μg/g), T = inlet temperature (°C), C = concentration of maltodextrin (wt. %).

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
