# Peer review of "Spray Drying of Blueberry Juice-Maltodextrin Mixtures: Evaluation of Processing Conditions on Content of Resveratrol"

_antioxidants, 2019, doi:10.3390/antiox8100437_

Round 1

Reviewer 1 Report

There are many points that make this manuscript not suitable for publication.

A major part of the data was already published in the polymers paper. (Polymers 2019, 11(2), 312) The merrit and novelty of this study is very low. Discussion of existing literature on spray drying of reveratrol is missing. The discussion lacks mechanistic thoughts. E.g. no explanations are given on the generally low retention The juice making process should have been more related to industrial juice making Details on the analytical parts are missing some minor point are commented in the pdf.

Author Response

RESPONSE TO REVIEWERS

The authors thank the reviewers for the valuable comments and changes suggested in the manuscript under review. All of these were considered and answered individually. We hope that these answers meet your expectations. As authors, we can say that the work was modified extensively, obtaining a more concise but better discussed work. Please find in the following sections, the answers for each of your comments and questions. The changes made in the text were highlighted with red font in the present document, the formal manuscript revised version (Microsoft Word file), and in the PDF file. The answers for the reviewer not included in the text, are in black font.

Reviewer 1:

Comment 1: (1) There are many points that make this manuscript not suitable for publication.

(2) A major part of the data was already published in the polymers paper. (Polymers 2019, 11(2), 312).

(3) The merit and novelty of this study is very low.

(4) Discussion of existing literature on spray drying of resveratrol is missing. The discussion lacks mechanistic thoughts. E.g. no explanations are given on the generally low retention (5) The juice making process should have been more related to industrial juice making.

(6) Details on the analytical parts are missing some minor point are commented in the pdf. peer-review-4984503.v1.pdf

Answer: (1) We appreciate your point of view and comments. We put our best effort in making all of these changes.

(2) As indicated by the reviewer, all the data results and discussion related with the physicochemical characterization and yield (previously reported in Polymers 2019, 11(2), 312)), were deleted from the text. Subsequently, the whole text was modified accordingly.

(3) After adding a short review of selected the works published in the topic of “optimization of processing conditions of blueberry juice and conservation of resveratrol”, the novelty of the work was more notorious, since those reported works help us to contrast the contribution of the present work.

(4) In the results and discussion section, some new references were added to improve the discussion level.

(5) The “industrial juice making” is out of the scope of the work, since we are not focused on the making of the juice, but on the research on the conservation of the antioxidants.

(6) Some technical details related to the identification and quantification of the resveratrol in the blueberry juice and spray dried samples, were added in the corresponding Experimental section.

Line 34: “produce” was changed by produces.

 Line 35: by was added before affecting.

 Line 38: The comma (,) was added after “However”.

 Line 40: Examples of other biomolecules were added as (i.e. proteins, lipids, and nucleic acids).

 Line 44: There are much more antioxidants in nature.

The sentence was modified as There are various types of antioxidants in nature, among them are found the flavonoids and stilbenoids structures such as quercetin 3-D-galactoside, resveratrol, myricetin, and kaempferol.

 Line 45: “Among” was replaced by Between.

 Line 46: The typo error was corrected as: 3, 4’, 5-trihydroxystilbene.

 Line 52: The Latin names style font was changed to italics as: Polygonum cuspidatum .

 Line 55: “sort” was changed by variety.

 Line 58: “developing” was replaced by to develop.

 Line 59: What is the meant of short biological life?

The following was added in the text: (i.e. rapid metabolism and elimination).

 Line 61: Please state the exact genus? V. corymbosum? V. myrtillus?

The genus of the blueberry fruit employed in the preparation of the juice was added as (V. corymbosum)

 Line 77: ANOVA is not really a method for optimization.

We agree with the comment of the reviewer. Thus, the complete sentence was corrected as Through the analysis of variance (ANOVA) and the response surface plots (RSP), they determined the effect of the processing conditions, found the optimal set of experimental variables for the drying of the juice and the conservation of quercetin 3-D-galactoside, and set the application limits of the maltodextrins based on the MWD.

 Line 80: please explain this. There is lots of literature on spray drying of resveratrol.

In accordance to the comment of the reviewer, we carried out a bibliographic search on this subject. Because of this search, we included several examples in the text that supported our work. Additionally, we deleted the sentence “Anyhow, to our best knowledge, these conditions still have not been reported for resveratrol”.  The following was added in the manuscript:

Although there is a large number of works published on the subject of conservation of resveratrol, few have addressed the issue of optimization of the processing conditions. For example, Lim, Ma & Dolan carried out a systematic experiment for testing the spray drying conditions of blueberry by-products. They found a yield of 94% when employing a ratio of blueberry solid to maltodextrin of 30:70, and an outlet temperature of 90 °C [1]. Jiménez-Aguilar et al. studied the spray drying conditions of blueberry and mesquite gum on the color and degradation of anthocyanins, and found a direct relation between color and concentration of anthocyanins [2]. Tatar Turan et al. carried out a study on the effect of the ultrasonic nozzle in the processing conditions of the spray drying of blueberry juice with maltodextrin and Arabic gum as carrier agent and coating material, respectively [3]. They reported the optimal drying conditions as inlet temperature of 125 °C, ultrasonic power of 9 W, and feed pump rate of 8%. Correia et al. encapsulated the polyphenolics in wild blueberry with different protein-food ingredients, by the two-way ANOVA and Tukey’s test; they found that soy protein produced the highest total phenolic content [4]. Darniadi, Ho & Murray, also used the ANOVA for comparing two methodologies in the drying of blueberry juice: foam-mat freeze-drying (FMFD) and spray drying. They concluded that the highest powder yield was achieved with the FMFD methodology and a ratio of maltodextrin to whey protein of 1.5 [5]. Shimojo et al. employed a 22 full factorial experimental design for evaluating the process parameters on the properties of resveratrol loaded in nanostructured lipid carriers [6]. Additionally, other works have also studied the drying processing conditions of alternative drying methodologies [7-9].

 Line 89: Why not V. myrtillus? much higher content in phenolics.

In addition to the commercial availability of the blueberry specie, the V. corymbosum is widely harvested and consumed in North America, while the V. myrtillus is harvested in Europe. According to Skrovankova et al. the chemical composition of berries varies with factors such as cultivar, variety, growing location and environmental, plan nutrition, ripeness time, harvest time and storage conditions (Skrovankova, S., Sumczynski, D., Mlcek, J., Jurikova, T., & Sochor, J. 2015. Bioactive compounds and antioxidant activity in different types of berries. International journal of molecular sciences, 16(10), 24673-24706)

 Line 97: excluding oxygen would have been more important.

In agreement with the comment of the reviewer, and accordance to Kalt, McDonald & Donner, oxygen has a marked effect on the degradation of anthocyanins, and total phenolic content of blueberry juice. Thus, keeping the blueberry juice in vials completely filled, eliminating the dissolved oxygen by nitrogen sparging, or by vacuum deaeration, are simple methods for reducing the degradation of the antioxidants (Kalt, W., McDonald, J. E., & Donner, H. (2000). Anthocyanins, phenolics, and antioxidant capacity of processed lowbush blueberry products. Journal of food science65(3), 390-393.). Experimentally, we decided to keep the blueberry juice in glass flasks closed with screw cap, as commonly is storage in the marketplace, i.e. in the fridge and away from direct sunlight. However, dissolved oxygen was not displaced by any of the methods aforementioned. Daily, for every set of spray drying experiments were employed fresh blueberry juice samples.

 Line 98: what are the specification for this commercial grade maltodextrin, and what is the difference to the others.

The main difference between the maltodextrins employed as carrying agents, was the dextrose equivalent, and their molecular weight distribution (MWD). The commercial grade (Mc) is a mixture of low molecular weight maltodextrins, but the supplier did not report its DE. After the physicochemical characterization reported by Saavedra-Leos et al. (2015), we found that the DE of this maltodextrin was 7. This value was included in the text as identified according to the dextrose equivalent (DE) as DE of 7 (commercial grade maltodextrin, Mc).

 Line 116: please include details on feed solution preparation.

The following description was added in the text: Blends of BJ-M were prepared by simply adding the necessary amount of maltodextrin in the juice, and mechanically mixing by constant stirring.

 Line 135: Which pH? Juice? Solvent?

The pH of 3 refers to the solution obtained from the extraction of resveratrol in dry samples. The details of the preparation of this solution were included in the text as The quantification of the antioxidant content in the dry samples was carried out by dissolving 0.5 g of powder in 0.5 ml of phosphoric acid (10% v/v in water), and 3 ml of methanol as extracting solvent. In order to maximize the extraction of the antioxidants the solution was stirred for 5 minutes, and left to rest 24 hours in darkness. Then solution was filtered in an Acrodisc filter (0.45 μm), and the filtered solution was diluted with 200 μL of methanol. A constant volume of 10 μL was injected in the HPLC instrument. Injections were carried out by triplicate.  

 Line 139: Please state details on compound identification.

The following resveratrol identification was added in the text: The calibration curves were constructed employing a resveratrol HPLC grade standard (>99.0%, Sigma-Aldrich). A stock solution of 1000 μg/ml, and several aliquots (0.01, 1, 5, 10, and 20 μg/ml) were prepared for the calibration curve. The calibration curves were prepared the same day of the injection in the HPLC. The elution time of the antioxidant was 2.5 min, while the mobile phase eluted at 0.92 min. The intensity of the resveratrol peak linearly increased with the concentration. The content of the antioxidant in the sample was determined from the comparison with the calibration curve.

 Line 160: Is this data from the previous publication?

Yes, it was data from the same experiment, but reported in a previous work. Anyhow, the complete section 3.1. Physical characterization was deleted from the text. The following explanation was included: From the 25 experiments, 18 runs were successfully spray-dried, while in the remaining seven runs the microstructure collapsed. Because the results for resveratrol are discussed herein, the corresponding results of the physicochemical characterization and yield can be consulted in the work previously reported for quercetin 3-D-galactoside [10], since both works are parts of the same experiment.

 Line 191: The yield data is EXACTLY the same, since it were the same experiments.

Same as previous question, the data, figures, tables and discussion related to the yield were deleted from the text.

 Line 194: Please explain. The retention of resveratrol is lower than that of quercetin

The corresponding explanation about the differences in the content of both antioxidants, was included in the last part of the result and discussion as 3.3. Effect of the chemical structure in the content of antioxidants.

 Line 237: Again same data as published before.

As mentioned before, the yield data and figures were deleted from the text.

 Line 238: What are A B C & D.

Although the figure caption for the yield was deleted, the figure caption of the response surface plots for the content of resveratrol was corrected as indicated by the reviewer. The following was added to the caption: Figure 2. RSP for the content of resveratrol as a function of the type of MX: A) Mc, B) M10, C) M20, and D) M40.

Reviewer 2 Report

The present manuscript displays a quite interesting topic, but could not be published in its current form, because there exist some major problems in this paper, which I will list as following.

Abstract:

Line 22: “The RSP…” –there should be no unexplained abbreviations in the abstract

Materials and Methods:

Line 108-109:” The response variables were the powder yield (Y), and the content of quercetin 3-D- galactoside retained (R).” - I think it was resveratrol instead of quercetin 3-D- galactoside.

Line 142-143: “where QP is the content of resveratrol in the dry powder (in ppm), and QJ is the content of the  antioxidant in the fresh juice (8.38 ppm).” - Were the results expressed in fresh juice calculate on a dry or fresh basis?

Table 2 and Figure 2 present the same results so I propose to delete the Figure 2

Line 202-206: “Thus, if F is equals to one, the independent variable has no effect, while if F>1 the independent variable has an effect and the effect is lager as F increases above 1. The p-value  represents the probability of a F-value large enough for influencing the experiment; if the p-value is  equal or lower than the significance level, then the assumption of the influence of the independent  variables on the response variable is correct. The probability of rejecting the previous assumption  even when is true, is given by the significance level.” – This is the methodology not results

Section 4. (Discussion ) is missing

Author Response

RESPONSE TO REVIEWERS

The authors thank the reviewers for the valuable comments and changes suggested in the manuscript under review. All of these were considered and answered individually. We hope that these answers meet your expectations. As authors, we can say that the work was modified extensively, obtaining a more concise but better discussed work. Please find in the following sections, the answers for each of your comments and questions. The changes made in the text were highlighted with red font in the present document, the formal manuscript revised version (Microsoft Word file), and in the PDF file. The answers for the reviewer not included in the text, are in black font.

Reviewer 2:

Comment: The present manuscript displays a quite interesting topic, but could not be published in its current form, because there exist some major problems in this paper, which I will list as following.

Answer: Authors appreciate the comments and suggestions made by the reviewer.

 Line 22: “The RSP…” –there should be no unexplained abbreviations in the abstract.

Abstract was modified according to the comments of the reviewer. This is the new abstract included in the text:

Resveratrol is an antioxidant abundant in red fruits, and one of the most powerful inhibiting reactive oxygen species (ROS) and oxidative stress (OS) produced by the human metabolism. The effect of the spray drying processing conditions of blueberry juice (BJ) and maltodextrin (MX) mixtures was studied on the content and retention of resveratrol. Quantitatively the analysis of variance (ANOVA) showed that concentration of MX was the main variable influencing the content of the antioxidant. The response surface plots (RSP) confirmed the application limits of maltodextrins based on their molecular weight, where the low molecular weight MX showed a better performance as carrying agents. The comparison of the resveratrol results versus those reported for a larger antioxidant molecule (quercetin 3-D-galactoside), qualitatively showed a higher influence of the number of available active sites rather than the stearic hindrance, on the chemical interactions.

Line 108-109: “The response variables were the powder yield (Y), and the content of quercetin 3-D- galactoside retained (R).” - I think it was resveratrol instead of quercetin 3-D- galactoside.

Totally agree with the comment. This error was corrected as indicated by the reviewer.

The response variable was the content of resveratrol.

Line 142-143: “where QP is the content of resveratrol in the dry powder (in ppm), and QJ is the content of the antioxidant in the fresh juice (8.38 ppm).” - Were the results expressed in fresh juice calculate on a dry or fresh basis?

Yes, the content of resveratrol was calculated from the fresh blueberry juice.

Line 195-196: Table 2 and Figure 2 present the same results so I propose to delete the Figure 2.

Figure 2 was deleted from the text. Table 2 was modified deleting the yield values.

Line 202-206: “Thus, if F is equals to one, the independent variable has no effect, while if F>1 the independent variable has an effect and the effect is lager as F increases above 1. The p-value represents the probability of a F-value large enough for influencing the experiment; if the p-value is  equal or lower than the significance level, then the assumption of the influence of the independent  variables on the response variable is correct. The probability of rejecting the previous assumption even when is true, is given by the significance level.” – This is the methodology not results

In accordance with the reviewer, the paragraph was removed from the results and discussion section to the corresponding experimental section 2.5. Statistical analysis as:

The effects of the three factors and their interactions were evaluated with the analysis of variance (ANOVA). In the ANOVA results, the F-value indicates the extent of the effect of the independent variable on the response variable. Thus, if F is equals to one, the independent variable has no effect, while if F>1 the independent variable has an effect and the effect is lager as F increases above 1. The p-value represents the probability of an F-value large enough for influencing the experiment; if the p-value is equal or lower than the significance level, then the assumption of the influence of the independent variables on the response variable is correct. The probability of rejecting the previous assumption even when is true, is given by the significance level.

Line 57: Section 4. (Discussion) is missing

The word Discussion was added to the section 3. Results. In the revised version of the manuscript, the section was renamed as 3. Results and discussion

Reviewer 3 Report

The manuscript is an interesting research about the application of spry drying on blueberry juice using maltodextrin as adjunct for resveratrol retention. The novelty of the research is not very high. The same authors cited similar works that reported the spry drying of juices and the use of maltodextrins in this operation. This work is part of a previously published one.  I suggest rewrite the manuscript without the parts (particularly data, tables and figures) already published.

Authors should check the entire manuscript, some typing mistakes occur along it. For example at line 45 there is a repetition of “such as”, at line 117 “m3” (3 must type at apex) and other small ones.

Abstract

Is really resveratrol the most powerful inhibiting ROS and OS? As well known, and the same authors reported in the introduction part (line 48), resveratrol is one of the most potent antioxidants against ROS and OS, not the most. Authors should change the assertion at line 18 of abstract. Authors report the abbreviations; I think they report also the complete name of MX, ROS, OS, RSP…

Introduction

Line 53 and 93 (and other parts along text). Authors should delete the year (2012 and 2019) as it is already reported by mean of the citations [9 and 21]. Also in other parts of the text authors reported both year and reference number. I think it is not completely correct. At line 84 the reference [21] is not reported.

Materials and methods

Lines 97-100. It is not clear the differences among the Commercial maltodextrin (Mc) and others (M10, M20 and M40). The M10, 20, 40 have a Destrose Equivalent index of 10, 20 and 40, but Mc was not reported a particular DE, or differ for other features? In the discussion part, authors reported the different degree of polymerization (lines 295-296), is this the only difference? The authors should better describe. “Content and retention of resveratrol”. Authors should briefly report how resveratrol was quantified in juice and dry powder, if they have used external or internal standard and the characters of standard used, mainly its purity. Statistical Analyses. Authors should report the software used to obtain statistical results and graphs.

Results and discussion

I suggest revise this part and don’t report data, table and figures presented in other manuscripts such as Polymers 2019, 11(2), 312. Figure 1. The legends and axis scale of SEM micrographs and XRD difractograms are not easy to read. If the authors could enlarge these parts, the readers could better appreciate it. Figure 2. The way of bar graph does not permit a well observation. Moreover, the same data are reported in table 2. I suggest delete figure 2, it is not necessary. Lines 201-203. When authors explain the ANOVA results, they affirm that have reported elsewhere (into text?) what F-value indicates. I think this could be reported into the methods described part, but it isn’t. I suggest modify the phrase deleting “As reported elsewhere”. Figure 3 and 4. Legend should explain better what the different graphs are. Letter A refer to Mc? And so on? 3. ANOVA and Response Surface Plots Analysis. The response of maltodextrin to yield and resveratrol retention could be influenced by other properties than degree of polymerization. Otherwise the opposite behavior respect temperature observed with Mc and M10 isn’t explained. The authors haven’t clarified this strange trend, where the lower polymerization degree showed the higher resveratrol content one at the minimum and the other at the maximum temperature used. 4. Effect of the Chemical Structure in the Content of Antioxidants. The discussion about the higher quercetin 3-galactoside in spry dry powder doesn’t take into account the higher content into juice of this compound respect resveratrol. As reported in the manuscript published on Polymers 2019, 11(2), 312, quercetin was 31.45 ppm, while resveratrol was 8.38 ppm: three times higher. The powder (trial 10) showed 1.98 microg/g of quercetin and 0.47 microg/g: four times higher. Without repetitions and a statistical analyses it is difficult define as different the behaviors. Figure 5 is useless. The structures of the reported molecules are well known.

Author Response

RESPONSE TO REVIEWERS

The authors thank the reviewers for the valuable comments and changes suggested in the manuscript under review. All of these were considered and answered individually. We hope that these answers meet your expectations. As authors, we can say that the work was modified extensively, obtaining a more concise but better discussed work. Please find in the following sections, the answers for each of your comments and questions. The changes made in the text were highlighted with red font in the present document, the formal manuscript revised version (Microsoft Word file), and in the PDF file. The answers for the reviewer not included in the text, are in black font.

Reviewer 3:

Comment: (1) The manuscript is an interesting research about the application of spry drying on blueberry juice using maltodextrin as adjunct for resveratrol retention. (2)The novelty of the research is not very high. (3) The same authors cited similar works that reported the spry drying of juices and the use of maltodextrins in this operation. (4) This work is part of a previously published one.  (5) I suggest rewrite the manuscript without the parts (particularly data, tables and figures) already published.

Answer: (1) Authors appreciate the comments and suggestions made by the reviewer.

(2) We believe that after the changes suggested by the three reviewers, the novelty of the work was more notorious. A short review of works reported in the same topic was included in the Introduction section.

(3) Yes, we cited several works related with the pray drying of juices. However, those works reported other antioxidants, drying process, and juices. The new references added in the form of the short review aforementioned, help us to contrast the novelty and contribution of the present work.

(4) Yes, this work is part of a experiment previously reported. Anyhow, the present work focused in effect of the spray drying processing conditions on the content and retention of resveratrol. The work previously published was focused on the content and retention of quercetin 3-D-galactoside.

(5) According with the suggestion of the reviewer, the manuscript was modified and corrected.  

Comment: Authors should check the entire manuscript, some typing mistakes occur along it. For example at line 45 there is a repetition of “such as”, at line 117 “m3” (3 must type at apex) and other small ones.

Answer: The text was revised and modified for correcting these typing mistakes.

Line 45: The error was corrected as m3/h.

Comment: (Abstract) Is really resveratrol the most powerful inhibiting ROS and OS? As well known, and the same authors reported in the introduction part (line 48), resveratrol is one of the most potent antioxidants against ROS and OS, not the most. Authors should change the assertion at line 18 of abstract. Authors report the abbreviations; I think they report also the complete name of MX, ROS, OS, RSP…

Answer: Abstract was modified as suggested by the reviewer. The new version of the abstract is:

Resveratrol is an antioxidant abundant in red fruits, and one of the most powerful inhibiting reactive oxygen species (ROS) and oxidative stress (OS) produced by the human metabolism. The effect of the spray drying processing conditions of blueberry juice (BJ) and maltodextrin (MX) mixtures was studied on the content and retention of resveratrol. Quantitatively the analysis of variance (ANOVA) showed that concentration of MX was the main variable influencing the content of the antioxidant. The response surface plots (RSP) confirmed the application limits of maltodextrins based on their molecular weight, where the low molecular weight MX showed a better performance as carrying agents. The comparison of the resveratrol results versus those reported for a larger antioxidant molecule (quercetin 3-D-galractoside), qualitatively showed a higher influence of the number of available active sites rather than the stearic hindrance, on the chemical interactions.

Line 53 and 93: (and other parts along text). Authors should delete the year (2012 and 2019) as it is already reported by mean of the citations [9 and 21]. Also in other parts of the text authors reported both year and reference number. I think it is not completely correct.

Answer: The whole document was modified as indicated by the reviewer. The publication year was delete from the references, and the citations left at the end of the sentence. The following is an example of the changes performed:

Recently, Saavedra-Leos et al. reported the use of a set of four maltodextrins as carrying agents in the spray drying of BJ [1].

Line 84: the reference [21] is not reported.

Answer: Authors appreciate this comment. All references were revised and corrected employing Refworks for this purpose.

Lines 97-100: It is not clear the differences among the Commercial maltodextrin (Mc) and others (M10, M20 and M40). The M10, 20, 40 have a Dextrose Equivalent index of 10, 20 and 40, but Mc was not reported a particular DE, or differ for other features?

Answer: The dextrose equivalent for the commercial maltodextrin was included in the text as DE of 7 (commercial grade maltodextrin, Mc).

Lines 295-296: In the discussion part, authors reported the different degree of polymerization, is this the only difference?

Answer: Saavedra-Leos et al. (Molecules 2015, 20, 21067-21081) previously reported the complete characterization of this set of maltodextrins. The main difference between these maltodextrins was the molecular weight distribution (MWD), and degree of polymerization. In consequence, the adsorption, thermal, and microstructural properties, and chemical interactions depend on the length and arrangement of the glucose polymeric chains.

Comment: The authors should better describe. “Content and retention of resveratrol”.

Answer: The following was included in the text:

Overall, the average concentration of resveratrol in the dried experiments diminished about 96%. This was calculated considering the initial concentration of resveratrol in the fresh BJ, and the determined content of the antioxidant in the dried sample. On the other hand, the content and retention values were relatively lower than those obtained for quercetin 3-D-galactoside [1]. Therefore, based on these observations, it is evident that resveratrol is less prone to interact chemically with the maltodextrins, being more exposed thermal degradation during the spray drying process.

Comment: Authors should briefly report how resveratrol was quantified in juice and dry powder, if they have used external or internal standard and the characters of standard used, mainly its purity.

Answer: In agreement with the reviewer, a more detailed description of the quantification of the resveratrol by HPLC was included in the text. The following paragraphs were added:

The quantification of the antioxidant content in the dry samples was carried out by dissolving 0.5 g of powder in 0.5 ml of phosphoric acid (10% v/v in water), and 3 ml of methanol as extracting solvent. In order to maximize the extraction of the antioxidants the solution was stirred for 5 minutes, and left to rest 24 hours in darkness. Then solution was filtered in an Acrodisc filter (0.45 μm), and the filtered solution was diluted with 200 μL of methanol. A constant volume of 10 μL was injected in the HPLC instrument. Injections were carried out by triplicate.

The calibration curves were constructed employing a resveratrol HPLC grade standard (>99.0%, Sigma-Aldrich). A stock solution of 1000 μg/ml, and several aliquots (0.01, 1, 5, 10, and 20 μg/ml) were prepared for the calibration curve. The calibration curves were prepared the same day of the injection in the HPLC. The elution time of the antioxidant was 2.5 min, while the mobile phase eluted at 0.92 min. The intensity of the resveratrol peak linearly increased with the concentration. The content of the antioxidant in the sample was determined from the comparison with the calibration curve.

Comment: Statistical Analyses. Authors should report the software used to obtain statistical results and graphs.

Answer: The following was added in the experimental section of the text:

The experimental design and RSP were carried out employing Matlab 2013a.

Comment: I suggest revise this part and don’t report data, table and figures presented in other manuscripts such as Polymers 2019, 11(2), 312.

Answer: The complete section was revised and modified as indicated by the reviewer. Figures and data were deleted from the text. Additionally, a new section was added.

Comment: Figure 1. The legends and axis scale of SEM micrographs and XRD difractograms are not easy to read. If the authors could enlarge these parts, the readers could better appreciate it.

Answer: Figure 1 was deleted from the text. The following legend was included in the text so that the reader can have a wider context about the results:

From the 25 experiments, 18 runs were successfully spray-dried, while in the remaining seven runs the microstructure collapsed. Because the results for resveratrol are discussed herein, the corresponding results of the physicochemical characterization and yield can be consulted in the work previously reported for quercetin 3-D-galactoside [1], since both works are parts of the same experiment.

Comment: Figure 2. The way of bar graph does not permit a well observation. Moreover, the same data are reported in table 2. I suggest delete figure 2, it is not necessary.

Answer: In agreement with the reviewer, Figure 2 (bar graph) was deleted from the text.

Lines 201-203: When authors explain the ANOVA results, they affirm that have reported elsewhere (into text?). I suggest modify the phrase deleting “As reported elsewhere”.

Answer: The affirmation aforementioned “as reported elsewhere” was deleted from the text.

Comment: what F-value indicates? I think this could be reported into the methods described part, but it isn’t.

Answer: The ANOVA results describing section (lines 201-207) was removed from the results and discussion, to the Experimental section 2.5. Statistical analysis.

Comment: Figure 3 and 4. Legend should explain better what the different graphs are. Letter A refer to Mc? And so on?

Answer: Figure 3 was deleted. Figure 4 caption was corrected as indicated by the reviewer, by adding the corresponding maltodextrin name.

Figure 2. RSP for the content of resveratrol as a function of the type of MX: A) Mc, B) M10, C) M20, and D) M40.

Comment: ANOVA and Response Surface Plots Analysis. The response of maltodextrin to yield and resveratrol retention could be influenced by other properties than degree of polymerization. Otherwise the opposite behavior respect temperature observed with Mc and M10 isn’t explained. The authors haven’t clarified this strange trend, where the lower polymerization degree showed the higher resveratrol content one at the minimum and the other at the maximum temperature used.

Answer: The following was added to the discussion section.

Indeed, the interaction between maltodextrin and the antioxidant (i.e. resveratrol or quercetin) may be affected by other variables beyond the degree of polymerization of the maltodextrin. Other variables that may affect this interaction into a greater or lesser extent are the volumetric flow of the solution injected into the dryer, the wet bulb temperature (i.e. relative humidity of the air), type of nozzle used (for example, regular versus ultrasonic), and adjuvant agents (i.e. soy protein, sodium alginate). However, in the present work, the ANOVA was focused on two spray drying variables (concentration of maltodextrin, and inlet temperature), and one categorical variable (type of maltodextrin). Through several works reported by the Saavedra-Leos [1-5], we have observed that in some juices such as orange and blueberries, their physicochemical properties vary with the differences in the type and distribution of molecular weight of the carrier. In these studies, we have attributed this behavior mainly to: (i) the differences in the molecular weight distribution of the carrier agent, (ii) the arrangement of the polymer chains (i.e. entangled or linear), and (iii) the type of molecule in these chains (for example, glucose for maltodextrin, or fructose for inulin). Additionally, according to Darniadi, Ho & Murray, when mixing low molecular weight sugars and high molecular active compounds, the active compound tend to segregate in some extent to the surface of the dried particle [6]. While this behavior reduces the sticking of the particles to the walls of the dryer, it also exposes the active ingredient to a faster degradation.

Comment: Effect of the Chemical Structure in the Content of Antioxidants. The discussion about the higher quercetin 3-galactoside in spry dry powder doesn’t take into account the higher content into juice of this compound respect resveratrol. As reported in the manuscript published on Polymers 2019, 11(2), 312, quercetin was 31.45 ppm, while resveratrol was 8.38 ppm: three times higher. The powder (trial 10) showed 1.98 microg/g of quercetin and 0.47 microg/g: four times higher. Without repetitions and a statistical analyses it is difficult define as different the behaviors.

Answer: First, we must mention that the reported values of content of resveratrol (and quercetin in the previous publication) represent the average value calculated from three repetitions. In this sense, the corresponding standard deviation value for the content of resveratrol was added into table 2. For all the trials or experiments, the ratio of the content of quercetin to resveratrol was bigger than 1, in the range of 2.77-7.82, with an average of 5.6. This indicates that content of quercetin was always higher. According to the observation done by the reviewer, the ratio of the antioxidants in the fresh juice was 3.75. When comparing the average value from the trials (5.6) against that from the fresh juice (3.75), is evident that the resveratrol is being lost during the drying process. On the other hand, the qualitative explanation given about the chemical interactions between the functional groups of the antioxidants and maltodextrin, clearly agree with what was discussed above.

Comment: Figure 5 is useless. The structures of the reported molecules are well known.

Answer: We partially agree with the comment of the reviewer. As he/she mentions, the chemical structures are well known; indeed, they can be found in Wikipedia. However, we consider that the schematic representation in the figure may help the reader to better understand the final part of the discussion. Additionally, we have found these structures reported in several works (Antioxidants 2017, 6, 73; Antioxidants 2019, 8, 244). For these reasons, the authors prefer to keep the figure in the text.

Round 2

Reviewer 2 Report

Accept

Author Response

Authors thank to reviewer 2 for the acceptance of the work.

Reviewer 3 Report

The authors have applied the suggestions. The manuscript in this version is well organized and report new data and discussions respect other researches published. An extensive review was inserted into Introduction part a new comments in the Discussion one.

Author Response

Authors thank to reviewer 3 for the comments.
